# Adjuvant Therapy for High-Risk Melanoma: An In-Depth Examination of the State of the Field

**DOI:** 10.3390/cancers15164125

**Published:** 2023-08-16

**Authors:** Islam Eljilany, Ella Castellano, Ahmad A. Tarhini

**Affiliations:** 1H. Lee Moffitt Cancer Center and Research Institute, Tampa, FL 33612, USA; 2Emory College of Arts and Sciences, Emory University, Atlanta, GA 30322, USA

**Keywords:** adjuvant, biomarkers, dabrafenib, immunotherapy, ipilimumab, melanoma, nivolumab, pembrolizumab, targeted therapy, trametinib

## Abstract

**Simple Summary:**

Patients with stage IIB-IV melanoma who have had surgery are recommended to receive systemic adjuvant therapy, intended to target the residual micro-metastatic disease and reduce the risk of melanoma relapse and death from melanoma. Multiple adjuvant therapy regimens have been tested over the past three decades leading to regulatory approvals. Due to significant improvements in relapse-free survival, programmed cell death protein 1 (PD-1) blockades and BRAF-MEK inhibitors (for BRAF mutant melanoma) are currently used as the standard-of-care adjuvant treatments for surgically resected stage III-IV melanoma. In the US, pembrolizumab has achieved regulatory approval as an adjuvant therapy for resected stage IIB-IIC melanoma, while nivolumab is currently under FDA review for this indication. This review examines melanoma’s phase III adjuvant treatment trials, focusing on the latest updates. It also reviews the role of biomarkers in potentially individualizing adjuvant therapy and summarizes the main limitations and future directions of adjuvant therapy options for high-risk melanoma.

**Abstract:**

The consideration of systemic adjuvant therapy is recommended for patients with stage IIB-IV melanoma who have undergone surgical resection due to a heightened risk of experiencing melanoma relapse and mortality from melanoma. Adjuvant therapy options tested over the past three decades include high-dose interferon-α, immune checkpoint inhibitors (pembrolizumab, nivolumab), targeted therapy (dabrafenib-trametinib for BRAF mutant melanoma), radiotherapy and chemotherapy. Most of these therapies have been demonstrated to enhance relapse-free survival (RFS) but with limited to no impact on overall survival (OS), as reported in randomized trials. In contemporary clinical practice, the adjuvant treatment approach for surgically resected stage III-IV melanoma has undergone a notable shift towards the utilization of nivolumab, pembrolizumab, and BRAF-MEK inhibitors, such as dabrafenib plus trametinib (specifically for BRAF mutant melanoma) due to the significant enhancements in RFS observed with these treatments. Pembrolizumab has obtained regulatory approval in the United States to treat resected stage IIB-IIC melanoma, while nivolumab is currently under review for the same indication. This review comprehensively analyzes completed phase III adjuvant therapy trials in adjuvant therapy. Additionally, it provides a summary of ongoing trials and an overview of the main challenges and future directions with adjuvant therapy.

## 1. Introduction

Melanoma arises from the uncontrolled proliferation of melanocytes in the skin [1,2]. In the United States, around 97,610 new cases of melanoma and 7990 melanoma-related deaths have been projected for 2023 [3]. The most common risk factors include UV exposure, having many moles, Caucasian descent, advanced age and familial predisposition [4]. In 2016, the American Joint Committee on Cancer (AJCC) developed the eighth edition of melanoma cancer staging, providing a consistent framework (TNM) for classification, prognosis assessment, and treatment planning [5]. It incorporates tumor thickness (T), ulceration, lymph node involvement (N) and metastasis (M), covering stages 0 to IV.

Genetic and molecular alterations are significant in melanoma pathogenesis [6]. Melanomas exhibit modifications in signaling pathways like the mitogen-activated protein kinase (MAPK) pathway and the phosphoinositol-3-kinase (PI3K)/AKT pathway, which regulate growth and proliferation [7]. The MAPK pathway is dysregulated in most melanomas due to mutations in the B-Raf proto-oncogene (BRAF) [8]. Additional oncogenic events involve other genes like *NRAS*, neurofibromin 1 (*NF1*), KIT proto-oncogene receptor tyrosine kinase (*KIT*), telomerase reverse transcriptase (*TERT*) gene, cyclin-dependent kinase inhibitor 2A (*CDKN2A*) gene, and phosphatase and tensin homolog (*PTEN*) gene, contributing to melanoma progression and metastasis [8,9,10,11,12,13,14,15]. Understanding these alterations aids in developing targeted therapies [6].

In the context of disease progression, melanoma often metastasizes to lymph nodes (LN) before spreading distantly [16,17,18]. Studies indicate a role for the lymphatic infiltration of neoplastic cells into the bloodstream, leading to hematogenous dissemination and metastasis [19,20,21,22,23,24]. However, distant metastases may arise from different clones than those within the examined LN [21,22]. It was suggested that melanoma cells traversing the lymphatic system have a higher propensity for dissemination and tumor formation than those spreading through the bloodstream [25]. Another important player in melanoma progression is the melanoma tumor microenvironment (TME) status, with diverse immune cell subsets that influence metastasis and prognosis [26,27]. Additionally, tumor-infiltrating lymphocytes (TILs) within the TME have been shown to impact prognosis and the likelihood of response to immune checkpoint inhibitors (ICIs) [28,29]. It has been shown that resistance to ICIs may involve tumor immune evasion mechanisms within the TME [30].

Managing individuals with primary melanoma involves the consideration of various factors and treatment options encompassing a wide range of interventions [31]. Standard procedures for localized melanoma include primary tumor-wide local excision (WLE) and sentinel LN (SLN) biopsy. Radiotherapy is viable for individuals with unresectable conditions or those who decline surgical treatment but have limited activity in this setting [32]. Systemic adjuvant therapies for high-risk melanoma, including interferon-α, have traditionally been used with limited uptake, owing to limited efficacy and significant toxicity. However, recent years have seen the emergence of innovative therapies resulting from a deepening understanding of tumor biology and host immunology. These have included targeted therapies for BRAF mutant melanoma and ICI immunotherapies that have transformed the current standard of care with significant improvements in clinical outcomes, safety profiles and quality of life (QoL) [33]. Therefore, for patients with resected stage III or stage IV melanoma, current systemic adjuvant therapy options include ICI therapy with pembrolizumab or nivolumab and targeted therapy with a dabrafenib–trametinib combination (for BRAF mutant melanoma). In addition, adjuvant pembrolizumab is an option for resected stage IIB and IIC melanoma where it has achieved regulatory approval, while nivolumab is currently under regulatory review for this indication. This article will review the role of these therapies as adjuvant therapy in melanoma through a literature search and comprehensive assessment of the published evidence on MEDLINE with PubMed interface (date of the last search: 31 May 2023) in order to accommodate the inclusion of relevant articles for this review (Table 1).

## 2. Adjuvant Therapies

By definition, adjuvant therapy is a supplementary treatment administered after the primary treatment, typically surgical intervention, to eliminate residual cancer and enhance the likelihood of disease control [31]. In melanoma, systemic adjuvant therapy is meant to target micro-metastatic disease that can be the source of future disease relapse. Figure 1 illustrates different types of adjuvant therapies that may apply to high-risk melanoma. 

### 2.1. Immunotherapy

The fundamental therapeutic approach of immunotherapy encompasses the hindrance of immune checkpoints, resulting in the activation of immune cells [34]. Among the potential treatments, cytokines like high-dose interferon-α-2b (HDI), inhibitors of cytotoxic T-lymphocyte antigen 4 (CTLA-4) and inhibitors of the programmed cell death-1 (PD-1) pathway have demonstrated significant benefits in randomized phase III trials, leading to regulatory approvals [35].

#### 2.1.1. Cytokine Immunotherapy with IFN-α

Prior to the emergence of ICIs, cytokines, including IFN-α and interleukin-2 (IL-2), were identified as some of the initial efficacious treatments for advanced metastatic melanoma. 

In 1995, HDI was approved by the FDA as an adjuvant therapy for resectable high-risk melanoma, marking its first approval for this indication [36]. IFN-α molecules are human protein-based agents that elicit the non-specific activation of the adaptive immune system through the Janus kinase (JAK)/signal transducer and activator of transcription (STAT) pathway, culminating in apoptosis induction in malignant cell lines. Research has demonstrated that it can trigger an increase in STAT1 expression while simultaneously leading to a decrease in STAT3 expression. This phenomenon may potentially rectify T-cell signaling abnormalities, which could lead to T-cell and dendritic cell infiltration into nodal metastases [37].

For the purpose of treating surgically resected AJCC 6th edition stage IIB and III melanoma, the Eastern Cooperative Oncology Group (ECOG) and Kirkwood et al. [38] analyzed the results of four multicenter randomized control trials (RCTs) [36,39,40,41], testing HDI for a total of 1916 patients. Among these, three phase III trials (E1684, E1690, and E1694) were conducted, in which patients were randomly assigned to either receive HDI, undergo observation (E1684, E1690) or receive the ganglioside GM2/keyhole limpet hemocyanin vaccine (GMK) (E1694). The studies demonstrated that HDI enhanced relapse-free survival (RFS) in the three trials and overall survival (OS) in the two trials, E1684 and E1694. A subsequent meta-analysis [42], comprising 14 RCTs, indicated that adjuvant therapy with IFN-α led to a statistically significant enhancement in both RFS and OS. Nevertheless, the clinical advantage was found to be low [hazard ratios (HRs) for disease recurrence and death were 0.82 and 0.89, respectively].

Subsequently, pegylated IFN-α2b was tested as adjuvant therapy for stage III melanoma and was approved by regulatory agencies, although its tolerability and frequent adverse effects limited its use [43,44]. Newer novel medicines with higher efficacy and tolerability have now replaced IFN-α2b as adjuvant therapy for high-risk melanoma patients [45].

#### 2.1.2. Anti-CTLA-4 Immunotherapy: Ipilimumab

Ipilimumab is a type of human monoclonal antibody that functions by inhibiting CTLA-4 and enhancing anti-tumor immune responses [46,47]. In 2011, the US FDA approved ipilimumab 3 mg/kg for the treatment of metastatic melanoma based on the results of the MDX-1020 trial [48]. 

Ipilimumab is the initial ICI authorized for adjuvant treatment among individuals diagnosed with resected stage III melanoma. In 2015, the FDA granted approval based on the findings of the EORTC 18071 trial [49]. The study compared the administration of ipilimumab 10 mg/kg (ipi10) to a placebo. In the ipi10 group, the median RFS was 26.1 months compared to 17.1 months in the placebo group. In 2019, after a median follow-up period of 6.9 years, ipilimumab exhibited enhanced outcomes in terms of RFS, distant metastasis-free survival (DMFS) and OS compared to the placebo [50]. A more recent study, U.S. Intergroup E1609, conducted by ECOG-ACRIN, investigated the efficacy of two distinct adjuvant ipilimumab doses, 3 mg/kg (ipi3) and ipi10 versus HDI [51]. The study met its primary endpoint, demonstrating that ipi3 resulted in a statistically significant enhancement in the OS rate against HDI. On the other hand, ipi10 failed to exhibit a significant advantage in terms of OS or RFS when compared to HDI or ipi3 with significantly lower treatment-related adverse events (trAEs) in favor of ipi3 vs. ipi10 (37% vs. 58%). Ipi 10 was not approved as an adjuvant therapy in Europe due to its association with a relatively high rate of trAEs, and the emergence of PD-1-inhibiting antibodies as a more efficacious and better tolerated adjuvant therapy for stage III melanoma patients [52].

#### 2.1.3. Anti-PD1 Immunotherapy with Nivolumab and Pembrolizumab

Both nivolumab and pembrolizumab are monoclonal immunoglobulin G4 (IgG4) antibodies that are fully human and target PD-1 [53,54]. Blocking PD-1 results in the reactivation of CD8+ T cells, thereby augmenting their quantity and efficacy. In 2017, the CheckMate-238 study [55] and, in 2019, the KEYNOTE-054 study [56] led to the approval of nivolumab and pembrolizumab, respectively, by the FDA as adjuvant treatments for patients with locoregionally advanced disease who have undergone complete surgical resection [57]. 

In the CheckMate-238 study [55], the administration of nivolumab yielded a marked improvement in RFS compared to ipi10 as an adjuvant therapy for stage IIIB-IV melanoma. Furthermore, nivolumab exhibited a superior tolerability profile, with fewer severe AEs (14.4% vs. 45.9%). These results were sustained in a recent 4-year follow-up of the Checkmate-238 trial [58], but the OS did not show any significant difference between the two groups. The findings of this trial led to the regulatory approval of adjuvant nivolumab [43,59].

KEYNOTE-054 [56] aimed to assess the effectiveness of adjuvant pembrolizumab therapy in individuals with clinical AJCC seventh edition stage III melanoma who had SLN metastases greater than 1 mm following resection and the completion of lymphadenectomy. The study findings indicated that the pembrolizumab arm had a significantly longer RFS and DMFS than the placebo arm. Furthermore, pembrolizumab did not appear to significantly reduce health-related QOL as compared to placebo [60]. 

As anti-PD1 adjuvant therapy effectiveness for stage III melanoma has been established, this has prompted its investigation in the treatment of resected high-risk stage IIB-C melanoma. In the KEYNOTE-716 study [61], patients with stage IIB-C melanoma, according to the AJCC eighth edition, were assigned randomly to receive either pembrolizumab or a placebo. The initial analysis demonstrated that pembrolizumab increased RFS significantly. Additionally, both treatment groups exhibited comparable measures of QOL without fatalities associated with the treatment. Thus, the FDA approved pembrolizumab as an adjuvant therapy in treating stage IIB or IIC melanoma after complete resection on 3 December 2021. In a later update of KEYNOTE-716 [61] at a median follow-up of 39.4 months until 4 January 2023, adjuvant pembrolizumab continued to demonstrate DMFS and RFS benefits compared to placebo, with a similar safety profile as reported before [62]. 

More recently, the results of the CheckMate-76K study (NCT04099251) designed to assess nivolumab’s efficacy versus placebo in patients with stage IIB-IIC cancer was presented at the Society For Melanoma Research (SMR) conference in 2022 (this is currently under review by the FDA) [63]. After a median follow-up of 15.8 months on nivolumab and 15.9 months on placebo, the study found that using nivolumab as an adjuvant therapy significantly improved RFS, with a reduction of 58% compared to placebo (HR 0.42; 95% CI 0.30–0.59; *p* < 0.0001). Not only that, but there was also a clinically meaningful improvement in DMFS with nivolumab compared to placebo (HR 0.47; 95% CI, 0.30–0.72). The 12-month rates of RFS for nivolumab were found to be 89% (95% CI; 86–92), whereas the placebo group exhibited a lower rate of 79% (95% CI; 74–84). The rtAEs rate was 15% among participants in the nivolumab treatment group, while it was 3% among those in the placebo treatment group.

The FDA approved the adjuvant use of ipilimumab, nivolumab and pembrolizumab for high-risk melanoma based on improved RFS as the primary endpoint of the pivotal clinical trials. On the other hand, ipilimumab exhibited an OS advantage for this particular indication when compared to the placebo in EORTC 18071 (secondary endpoint) and as compared to HDI in E1609 (primary endpoint). Furthermore, OS was evaluated in the phase III SWOG S1404 study [64], which incorporated OS as one of its co-primary endpoints. The study randomly sampled patients into two groups, one receiving pembrolizumab and the other receiving standard-of-care adjuvant immunotherapy consisting of ipi10 (primarily) and HDI. The conclusive evaluation of SWOG S1404 revealed that RFS was significantly improved. However, no improvement in the OS was observed.

#### 2.1.4. Combined Checkpoint-Inhibitor Therapy with Nivolumab plus Ipilimumab

The IMMUNED trial [65] was a phase II trial investigating whether the combination of ipilimumab and nivolumab would yield better outcomes than nivolumab or observation alone in patients with surgically resected stage IV melanoma who had undergone surgery or radiation and were found to have no evidence of disease. One hundred and sixty-seven patients were assigned randomly in a 1:1:1 ratio to nivolumab 1 mg/kg and ipilimumab 3 mg/kg every 3 weeks for four doses followed by nivolumab maintenance, or nivolumab 3 mg/kg every 2 weeks and ipilimumab placebo, or double placebo. At a median follow-up of 28.4 months, the median RFS was not reached in the combination arm (HR 0.23, 97·5% CI 0·12–0·45; *p*-value < 0.0001 compared to placebo), whereas it was 12.4 months for the nivolumab monotherapy arm (HR 0.56, 97·5% CI 0·33–0·94; *p*-value = 0.011 compared to placebo) and 6.54 months in the placebo arm. Comparable levels of toxicity were observed across all treatment groups, consistent with prior reports, and no fatalities were attributed to treatment. 

The CheckMate-915 study [66] is the most recent international multicenter trial that assessed the efficacy of the nivolumab and ipilimumab combination as compared to nivolumab monotherapy in 1844 participants, encompassing individuals with resected AJCC 8th edition stage IIIB-D and IV. The study represents the initial adjuvant trial for stage III melanoma that did not mandate complete lymph node dissection with a new combination therapy regimen. After a follow-up period of around 24 months, there was no significant difference in RFS between the two treatment arms, which was 64.6% for the combination treatment vs. 63.2% for nivolumab (*p* = 0.269), while the combination treatment group exhibited a higher level of toxicity. 

## 3. Targeted Therapy

### 3.1. Introduction to Targeted Therapy

In recent years, the clinical utilization of combined BRAF/MEK inhibitors has significantly improved the therapeutic landscape for patients with metastatic melanoma. Approximately 40–60% of melanoma patients possess a melanoma BRAF mutation, with the most frequent mutation occurring at codon 600 (BRAF V600E) [67]. The genetic alteration leads to the continuous activation of the RAF–MEK–ERK signaling cascade, which plays a pivotal role in the pathogenesis and progression of melanoma [68]. 

### 3.2. Adjuvant Targeted Therapy Studies in Melanoma

The efficacy of vemurafenib, the initial BRAF-targeting drug, as an adjuvant therapy was tested in the BRIM8 trial [69]. The BRIM8 study randomized patients with stage IIC-IIIB (cohort 1) or IIIC (cohort 2) and a melanoma-positive BRAF V600 mutation to receive either vemurafenib or placebo. The study followed a hierarchical analytic approach in testing the primary endpoint of disease-free survival (DFS) separately in each cohort, with cohort 2 being prespecified before cohort 1. The study’s results revealed that in the second cohort (stage IIIC), the median DFS was 23.1 months with vemurafenib versus 15.4 months with placebo, HR 0.80, 95% CI (0.54–1.18; *p* = 0.26). Moreover, the study demonstrated that the median DFS of the first cohort was not reached in the vemurafenib group versus 36.9 months in the placebo groups, HR 0.54, 95% CI (0.37–0.78; *p* = 0.0010). This outcome in cohort 1 was not considered significant, owing to the prespecified hierarchical prerequisite.

COMBI-AD [70,71] evaluated the efficacy of adjuvant targeted therapies, dabrafenib (BRAF inhibitor) and trametinib (MEK inhibitor) combination therapy in patients with stage III melanoma and BRAF V600 mutation. The estimated 3-year rate of RFS was significantly higher in the combination therapy group compared to the placebo group (RFS_combination_ = 58%, RFS_placebo_ = 39%, *p* < 0.001) as was the 3-year OS (OS_combination_ = 86%, OS_placebo_ = 77%, *p* = 0.0006). The DMFS and freedom from relapse were also higher. Finally, the safety profile of dabrafenib plus trametinib was similar to what was reported in patients with metastatic melanoma. After five years, the RFS and DMFS were persistently higher in the dabrafenib plus trametinib group versus placebo. An estimated cure rate analysis also supported potential OS benefits [71].
cancers-15-04125-t001_Table 1Table 1Completed immune checkpoint blockade and targeted therapy adjuvant phase III trials in melanoma.TrialNStageTrial GroupsRegimenRFS (%) (HR; *p*-Value)OS (%)(HR; *p*-Value)ConclusionEORTC-18071 [49,50]1211AJCC 7th IIIA, IIIB, IIICIpi 10 vs. placeboIpi 10 mg/kg every three weeks for four doses, then tri-monthly for up to 3 years* 26.1 vs. 17.1 months (0.76; *p* < 0.001)60.0 vs. 51.3 (0.73; *p* = 0.002)There was a significant increase in RFS, DMFS and OS following adjuvant treatment with ipilimumab.Intergroup-E1609 [51]1673AJCC 7th IIIB, IIIC, IVIpi 3 mg/kg vs. HDIIpi 10 mg/kg vs. HDIIpi 3 mg/kg or 10 mg/kg IV tri-weekly for four doses, followed by the same dose every 12 weeks for up to four dosesORIV HDI at 20 Mu/m^2^ of body surface area/day for five days/week for four weeks, followed by 10 Mu/m^2^/day subcutaneously every other day, three days per week, for forty-eight weeks4.5 vs. 2.5 years (0.85; *p* = 0.065) * 3.9 vs. 2.4 years (0.84; NS) *72 vs.67 (0.78; *p* = 0.044)70 vs.65 (0.88; NS)There was a significant survival benefit with ipi3 compared to HDI (primary endpoint), while ipi10 was more toxic and did not improve survival.Checkmate-238 [55,58]906AJCC 7th IIIB, IIIC, IVNivolumab vs. ipi 10 mg/kgNivolumab IV (3 mg/kg every 2 weeks) or ipi IV (10 mg/kg tri-weekly for four doses and then every 12 weeks) for up to one year51.7 vs. 41.2 (0.71; *p* = 0.0003)77.9 vs. 76.6 (0.87; *p* = 0.31)There was a sustained RFS benefit with nivolumab compared to ipilimumab.KEYNOTE-054 [56]1019AJCC 7th IIIA-CPembrolizumab vs. placeboPembrolizumab 200 mg IV/tri-weekly for 18 doses75.4 vs. 61.0 (0.57; *p* < 0.001)N/APembrolizumab significantly prolonged RFS over placebo without new adverse events.Keynote-716 ^¥^ [61]976AJCC 8th IIB, IICPembrolizumab vs. placeboPembrolizumab 200 mg tri-weekly for 17 cycles. In pediatric patients, 2 mg/kg Pembrolizumab89 vs. 83 (0.65; *p* = 0.0066N/ACompared with a placebo, pembrolizumab significantly reduced disease recurrence or death for one year but not for two years.CheckMate-76K [63]790IIB, IICNivolumab vs. placebo
89 vs. 79 (0.42; *p* ≤ 0.0001)N/ANivolumab was an effective adjuvant treatment option in resected stage IIB/C melanoma.SWOG S1404 [64]1378AJCC 7th IIIA-C, IVPembrolizumab vs. standard of care (ipi 10 mg/kg or HDIPembrolizumab 200 mg IV tri-weekly for a yearvs.Ipi 10 mg/kg IV tri-weekly for four doses, then every 12 weeks for up to three years ORHDI 20 MU/m2 IV for 1–5 days/week for four weeks, followed by 10 MU/m2/d SC days 1, 3 and 5, for a year57.6 vs. 38.6 (0.76; *p* = 0.00278.3 vs. 59.3 (0.82; NS)Pembrolizumab increased RFS but not OS compared to HDI or ipilimumab as an adjuvant.Checkmate-915 * [66]1844AJCC 8th IIIB-D, IVNivolumab and ipi 1 mg/kg vs. NivolumabNivolumab 240 mg bi-weekly plus ipi 10 mg/kg every six weeks for up to 1 year ORnivolumab 480 mg every 4 weeks for up to 1 year64.6 vs. 63.2 (0.92; NS)N/ARFS was not improved with nivolumab 240 mg + ipilimumab 1 mg/kg vs. nivolumab 480 mg.BRIM-8 [69]498AJCC 7th IIC-IIIB, IIICVemurafenib vs. placeboVemurafenib 960 mg or placebo BID for 1 yearIIC-IIIB: DFS, NR vs. 36.9 months (0.54; NS)IIIC: DFS, 23.1 vs.15.4 months (0.80, NS)IIIC: 79.6 vs. 79.1IIC-IIIB: 89.8 vs. 82.2The outcome in cohort 1 was not considered significant owing to the prespecified hierarchical prerequisite.COMBI-AD [70,71]870AJCC 7th IIIA-IIICDabrafenib + trametinib vs. placeboDabrafenib 150 mg BID + trametinib 2 mg QID or placebo up to 1 year* NR vs. 16.6 (0.51; *p* < 0.001)97.0 vs. 94.0 at 1 year, 91.0 vs. 83.0 at 2 years, and 86.0 vs. 77.0 at 3 years (0.57; *p* = 0.0006)Dabrafenib and trametinib as adjuvant therapy combinations in patients with BRAF V600E or V600K mutations resulted in improved RFS versus placebo.* Median RFS. ^¥^ We did not include part 2 results as they have not been reported yet. AJCC: American Joint Committee on Cancer; DFS: disease-free survival; DMFS: distant metastasis-free survival; HDI: high dose interferon; HR: hazard ratio; INF: interferon; ipi: ipilimumab; IV: intravenous; MU: million units; N/A: not available; N: the number of patients; NR: not reached; NS: nonsignificant; OS: overall survival; *p*: *p*-value; PFS: progression-free survival; RFS: recurrence/relapse-free survival; S.C: subcutaneous; vs.: versus.


### 3.3. Adjuvant Radiation Therapy

Research on adjuvant radiation therapy for regionally advanced melanoma has shown a decrease in the likelihood of local relapse but has not yielded significant advantages in RFS or OS. In practice, adjuvant radiotherapy is rarely used in high-risk melanoma and is limited to cases with positive surgical margins [72].

## 4. Discussion

### 4.1. Challenges and Limitations

#### 4.1.1. Identifying Candidates for Adjuvant Therapy

The disease stage and BRAF mutation status are the key variables to consider when it comes to systemic adjuvant therapy. For resected stage III melanoma, adjuvant immunotherapy with an anti-PD-1 agent is recommended for patients with BRAF wild-type disease. Patients with BRAF V600 E/K mutant melanoma have the option of anti-PD-1 immunotherapy or the supplementary alternative of dabrafenib and trametinib as oral tablets. For the latter group, factors to consider may include the distinct side-effect profiles associated with the two therapeutic modalities. Importantly, patients with active autoimmune diseases are at an increased risk for trAEs; therefore, immunotherapy is generally avoided, and dabrafenib and trametinib are usually preferred in this setting. Further, immunotherapy is typically preferred in patients with a history of heart arrhythmias, given the potential cardiac risks with BRAF-MEK inhibitors [73]. 

#### 4.1.2. Adjuvant Therapy Toxicity

The threshold for acceptable toxicity levels ought to be lower for patients undergoing adjuvant therapy than those receiving treatment for metastatic disease, given that a subset of patients would have already achieved a cure through surgical intervention and do not benefit from systemic therapy. Furthermore, AEs stemming from immunotherapy have the potential to be severe and, in certain cases, irreversible [70]. This includes requiring life-long hormone replacement, with rates of 10% for thyroid replacement (hypothyroidism), 1% for corticosteroid replacement (adrenal insufficiency) and 0.5–1% for insulin replacement (diabetes). Since the anti-tumor efficacy of ICIs promotes various immune-related adverse events (irAEs), the paucity of knowledge regarding the safety of ICIs in patients with cancer and preexisting autoimmune disease represents a significant challenge. This is mainly due to the fact that patients with a history of autoimmunity were not represented in ICIs clinical trials [74]. This prompted retrospective research efforts that examine the safety and tolerability of ICIs in patients with autoimmune diseases. For example, according to the 2018 systematic review findings [74], which included patients with multiple malignancies (84% were patients with melanoma) with active or inactive autoimmune diseases, three-quarters reported an aggravation of preexisting autoimmune disease, irAEs, or both. Interestingly, there were no differences in adverse events between patients with active and inactive disease. Moreover, flares and irAEs in patients with autoimmune disease on ICIs can often be managed without the need for treatment discontinuation, while some events can be severe and fatal. In the largest series study [75], which looked back at the treatment with ipilimumab in 30 patients with melanoma and preexisting autoimmune disease, it was found that around a quarter of patients experienced flares of the underlying autoimmunity and a third experienced new irAEs, including death in one patient. Subsequently, the same group recently published a retrospective assessment of 52 patients from the same cohort treated with an anti-PD-1 drug, with more than half (54%) having previously received ipilimumab. Specifically, the underlying autoimmune diseases flared in 38% of patients, de novo irAEs occurred in 29% and ICIs were discontinued in 12%. As expected, those with active preexisting autoimmune diseases experienced more flares than those with inactive diseases [76]. Finally, the most recent systematic review and meta-analysis [77] specifically looked at patients with pre-existing inflammatory bowel disease (IBD), and approximately 40% of the examined patients with melanoma experienced IBD relapse with ICIs, with most relapsing patients requiring corticosteroids and a third requiring biologics such as tumor necrosis factor (TNF) blockers.

In the context of targeted therapy, instances of enduring toxicities are infrequent. At the same time, it was observed that 26% of patients who participated in the COMBI-AD adjuvant clinical trial experienced severe events that resulted in the cessation of treatment. Current research is ongoing to identify dependable biomarkers to accurately forecast patients at an elevated risk of experiencing toxicity [70].

#### 4.1.3. Economic Impact of Adjuvant Therapy

In 2017, Tarhini and colleagues examined the associated cost with locoregional and distant recurrence episodes following the first melanoma surgery [78]. From a healthcare payer perspective, they found that for patients between 2008 and 2017, the average overall healthcare expenditure for all causes during a median follow-up period of 23.1 months was found to be USD 2645 per patient per month (PPPM) for locoregional recurrence episodes and USD 12,940 PPPM for distant recurrence episodes. The question here was whether new adjuvant therapies, including ICI, would be cost-effective considering the cost of managing disease recurrence. 

The justification for the duration of adjuvant therapy remains uncertain from both a clinical and biological perspective, but studies have followed the original adjuvant IFN studies that mostly treated patients for one year. In 2018, Stav et al. [79] tried to estimate the cost of of adjuvant immunotherapy as examined in the clinical trials that focused on PD-1, PD-L1 and CTLA-4 blockers. According to their analyses, the estimated cost per patient for a one-year treatment of melanoma with nivolumab was USD 165,000, translated to USD 13,750 per month. On the other hand, the cost of a three-year treatment with ipilimumab is estimated to exceed USD 1,850,000. Assuming complete adoption, the total target population in the United States incurs an estimated annual expense of USD 1.15 billion for adjuvant therapy with nivolumab for melanoma. 

When evaluating targeted therapies and checkpoint inhibitors, it is important to deliberate expenses beyond the drugs’ monetary value. These expenses may include monitoring expenses such as laboratory tests, delivery expenses and the costs associated with AEs resulting from the drugs. This may potentially also include the costs of treating recurrent disease with modern systemic therapeutics, the costs of work lost from disease recurrence and the costs incurred by family members caring for patients with recurrent disease, among others. 

### 4.2. Potential Future Directions in Adjuvant Therapy for Melanoma

Despite significant advancements in managing patients with melanoma, several uncertainties and controversies persist, and there is a pressing need for more efficacious treatment options. 

#### 4.2.1. Personalized Medicine

There exist several therapeutic alternatives at present for the treatment of advanced melanoma. The presence of an activating mutation in BRAF suggests potential advantages in utilizing targeted therapy involving BRAF-MEK for melanoma cases with such mutations. However, identifying dependable clinical-grade biomarkers for immunotherapy remains to be established. 

The investigation of programmed death-ligand 1 (PD-L1), a probable indicator expressed on tumor cells as a response to anti-PD1, is a significant area of interest in studying melanoma. Research has demonstrated enhanced PFS and OS rates in advanced melanoma patients who belong to the PD-L1-positive subgroup compared to those in the PD-L1-negative subgroup [80]. Certain patients exhibit low levels of PD-L1 expression but respond positively to ICIs and vice versa; this could complicate the predictive value of PD-L1 expression. Hence, the expression of PD-L1 currently lacks sufficient biomarker potential in melanoma to identify eligible candidates for anti-PD-1 therapy and judicious patients who would obtain optimal benefits from either monotherapy or combination immunotherapy [80].

A considerable amount of research on TME has identified that TILs are associated with improved patient survival in melanoma tumors [81]. Some studies have reported increased densities of CD8+ T-cell infiltrates in serial tumor biopsies of patients with metastatic melanoma who received ipilimumab and pembrolizumab and exhibited a positive response to the therapy [82,83]. The results are intriguing; however, the baseline CD8+ T-cell density exhibited a degree of overlap between patients who experienced a positive response and those who experienced disease progression. This overlap restricts the determination of an absolute value or range that could serve as a biomarker with clinical utility [84].

Another emerging biomarker is tumor mutational burden (TMB) which quantifies the number of somatic mutations per mega-base of genomic DNA. It has been proven that it is correlated with the number of neoantigens generated by the tumor and the response of patients with melanoma to ICIs [85,86]. However, a considerable proportion of patients who exhibit elevated TMB levels do not exhibit a response to anti-CTLA-4 or anti-PD-1 therapies. Subsequently, combining TMB with IFN-γ can potentially distinguish individuals who may derive therapeutic advantages. Exploratory biomarker analyses of the COMBI-AD clinical trial [70,71] concluded that individuals with low TMB and high IFN-γ tumors experienced the most significant advantage with dabrafenib plus trametinib compared to the placebo. 

Circulating tumor DNA (ctDNA) could differentiate the prognosis of patients across AJCC stages and within the same stage groups. In a 2018 study, where researchers evaluated the efficacy of anti-vascular endothelial growth factor (VEGF) bevacizumab in resected high-risk stage II and III melanoma patients randomly selected from the AVAST-M trial [87] it was revealed that patients who exhibited ctDNA after receiving curative surgery experienced a reduction in their disease-free interval (DFI) and OS compared to those who did not exhibit ctDNA [88].

After one year of the previous study, another research group examined the efficacy of ctDNA in patients with melanoma stage IIIB-D who did not receive systemic adjuvant therapy [89]. The study found that patients who exhibited detectable ctDNA had a median RFS significantly lower than 5.7 months compared to that observed in patients without detectable ctDNA. 

Lastly, in high-risk resected melanoma patients, serum LDH and S100B levels could predict prognosis [90,91]. In this context, according to Tarhini et al. [90], baseline S100B levels correlated significantly with RFS and OS, with higher values indicating a worse outcome. 

A recent preprint study by Wheeler et al. [92] analyzed the tumor microbiome and its correlation with genes and pathways in patients with metastatic melanoma who underwent immunotherapy. The study identified several microorganisms linked to the response to immunotherapy and immune-system-related gene expression. Moreover, CA209-76K is a sub-study [93] of the ongoing Checkmate-76K trial (NCT04099251). The study revealed that higher levels of IFN-γ signature [HR 0.59, (95% CI 0.41–0.86)], TMB [HR 0.66, (95% CI 0.49–0.90)], CD8+ T cells [HR 0.60, (95% CI 0.39–0.90)] and lower C-reactive protein (CRP) levels [HR 1.37, (95% CI 1.00–1.88)] were associated with improved RFS in patients treated with nivolumab. Also, nivolumab prolonged RFS over placebo across all biomarkers. 

#### 4.2.2. Phase III Trials in Development for Adjuvant Therapy

The clinical trial RELATIVITY-098 (NCT05002569) is currently in progress. It is aiming to compare the efficacy of a fixed-dose combination of relatlimab [lymphocyte-activation gene 3 (LAG-3)] antagonist and nivolumab versus nivolumab monotherapy in patients diagnosed with stage III or IV melanoma who have undergone a complete removal of their tumor(s). Additionally, KEYVIBE-010 (NCT05665595) is a new study that is assessing the RFS in patients with adjuvant pembrolizumab + vibostolimab (MK-7684A) as anti-TIGIT (T cell immunoglobulin and ITIM domain) compared to pembrolizumab alone (Table 2). Lastly, an ongoing seven-year trial with ClinicalTrials.gov ID: NCT05608291 is comparing fianlimab as anti-LAG-3 plus cemiplimab (anti-PD-1) to pembrolizumab as adjuvant therapy in completely resected high-risk melanoma patients.
cancers-15-04125-t002_Table 2Table 2Phase III clinical trials ongoing for adjuvant therapy.TrialNStageTrial ArmsPrimary OutcomeClinicaltrials.gov IDComparing fianlimab + cemiplimab to pembrolizumab in patients with completely resected high-risk melanoma1530IIC, III, IVFianlimab + Cemiplimab vs. PembrolizumabRFSNCT05608291KEYVIBE-0101650IB, IIC, III and IVPembrolizumab + Vibostolimab (MK-7684A) vs. PembrolizumabRFSNCT05665595RELATIVITY-0981050IIIA (>1 mm), B, C, D, IVRelatlimab + Nivolumab vs. NivolumabRFSNCT05002569V940-0011089II-IVmRNA-4157 + Pembrolizumab vs. Placebo + PembrolizumabRFSNCT05933577N: number of patients; RFS: recurrence/relapse-free survival.


#### 4.2.3. Vaccines and Cell-Based Therapies

Cancer vaccines, like the 6-Melanoma Helper Peptide (6-MHP), activate the immune system to target specific cancer antigens. Despite limited efficacy in advanced metastatic disease, trials of 6-MHP in patients with resected stage IV melanoma have shown significantly increased median survival compared to controls [94]. The phase II trial with ClinicalTrials.gov ID (NCT03897881), known as the Moderna mRNA-4157-P201 trial or KEYNOTE-942 [95], aimed to assess the effectiveness of pembrolizumab in combination with a personalized cancer vaccine, mRNA-4157/V940, compared to pembrolizumab alone as adjuvant therapy in patients who have undergone a complete resection of high-risk melanoma (Table 2). The primary data of one and half years of follow-up were presented by Khattak A et al. at the American Association for Cancer Research^®^ (ACCR) annual meeting this year and showed that mRNA-4157/V940 + pembrolizumab showed a significant 44% reduction in the risk of recurrence or death in patients than pembrolizumab alone [HR = 0.561 (95% CI 0.309–1.017), *p* = 0.0266]. Similarly, DMFS was lower in the pembrolizumab group than mRNA-4157/V940 + pembrolizumab [HR = 0.347 (95% CI 0.145–0.828), *p* = 0.0063]. Interestingly, the trAEs number was similar among both groups. As a next step, a phase III trial is planned, which will be an active-comparator-controlled clinical trial called “V940-001” (NCT05933577) among participants with high-risk stage II-IV melanoma (Table 2).

The outcomes of clinical trials for melanoma and cancer vaccines, in general, have been largely unsatisfactory in cases of advanced disease, with ORRs < 5% [96]. However, emerging data with mRNA individualized neoantigen therapy are very promising and have the potential to make a significant difference in the adjuvant setting. 

#### 4.2.4. Adjuvant Therapy and Survival Impact

The impact of adjuvant anti-PD-1 immunotherapy on OS in high-risk melanoma patients after surgical removal is unclear. Despite improvements in RFS in major randomized trials like KEYNOTE-054 [56], KEYNOTE-716 [61], CheckMate-238 [55] and S1404 [64], no significant OS benefits have been demonstrated. This raises questions in managing lower-risk patients, like stage IIIA, who may derive less clear and no OS benefits. Important to note that patients with stage IIIA melanoma and an SLN tumor burden of <1 mm were excluded from the ICI and COMBI-AD adjuvant trials. These considerations in relation to OS benefit also apply when considering adjuvant therapy for stage IIB and IIC melanoma where the OS benefits are not clearly supported. Future phase III trials should consider OS a co-primary endpoint and continue enabling crossover at relapse with close long-term follow-up for relapse, salvage therapy and overall survival. In addition, utilizing prognostic biomarkers for better patient risk stratification is important and should be incorporated prospectively in clinical trials. More resources should be invested in identifying biomarkers that predict who would benefit most from immunotherapy and targeted therapy.

## 5. Summary, Recommendations and Conclusions

The choice of adjuvant treatment for individuals with surgically excised cutaneous melanoma depends on multiple determinants, including tumor stage, BRAF mutation status, the probability of relapse, comorbid conditions and patient wishes. Patients presenting with high-risk BEAF WT melanoma are typically prescribed adjuvant anti-PD-1 immunotherapy in the absence of limiting comorbid conditions such as symptomatic autoimmunity. Those with BRAF-mutant melanoma have the options of anti-PD-1 monotherapy or dabrafenib–trametinib where the choice depends on patients wishes for oral agents versus intravenous therapy, the adverse event profiles, and comorbid conditions. Despite adjuvant ipilimumab showcasing favorable overall survival benefits when compared to placebo (EORTC 18071) [49] and to interferon-α (E1609) [51], it is inferior to anti-PD1 with regard to RFS and presents heightened toxicity risks. Therefore, it is also important to highlight that the anti-PD-1 antibodies, specifically nivolumab and pembrolizumab, exhibit significant superiority over ipilimumab in both RFS and safety profiles. The employment of pembrolizumab as an adjuvant treatment has obtained approval for managing surgically resected stage IIB and IIC melanoma, as corroborated by the latest evidence from the Keynote-716 study [61]. The combination of dabrafenib and trametinib has proven effective as an adjuvant treatment for stage III patients with melanoma that is BRAF V600E/K mutant. Upcoming trials should integrate the analysis of predictive and prognostic biomarkers into their clinical trial frameworks.

## Figures and Tables

**Figure 1 cancers-15-04125-f001:**
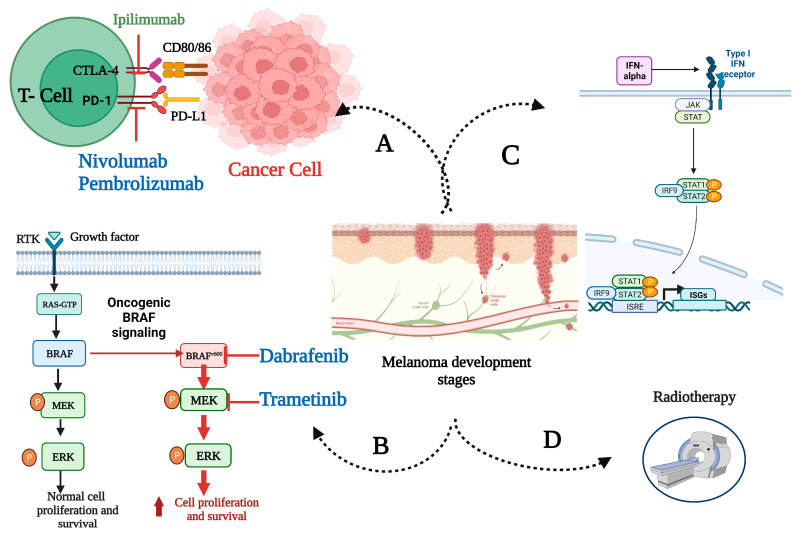
Molecular mechanisms underlying approved adjuvant therapies for high-risk melanoma. After the primary surgical management of locoregionally advanced disease, adjuvant therapies are administered as one or more of the following: (A) immune checkpoint inhibitors; (B) BRAF/MEK-directed targeted therapy for BRAF mutant melanoma; (C) interferon-alpha; (D) radiotherapy (in rare, selected cases). Immune checkpoint inhibitors boost melanoma immune responses in peripheral lymph nodes and the tumor microenvironment (TME). Ipilimumab, a CTLA-4 antibody, affects lymph node T cell priming and activation. Anti-PD-1/effector cell antibodies nivolumab and pembrolizumab restore the activity of T-cells in the TME. Targeted treatments, dabrafenib and trametinib, inhibit the mutant RAF–MEK–ERK signaling cascade, disrupting melanoma cell proliferation and differentiation. Created with BioRender.com. Abbreviations: BRAF^v600^, mutation occurring at codon 600 of BRAF gene; CTLA-4, cytotoxic T-lymphocyte-associated protein 4; ERK, extracellular signal-regulated kinase; IFN, interferon; IRF9, IFN regulatory factor 9; ISGs, IFN-stimulated gene factors; ISRE, IFN-stimulated response element; JAK, Janus kinase; PD-1, programmed cell death protein 1; PD-L1, programmed cell death-ligand 1; RTK, receptor tyrosine kinases; STAT, signal transducer and activator of transcription.

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
