# Peer review of "Adjuvant Therapy for High-Risk Melanoma: An In-Depth Examination of the State of the Field"

_cancers, 2023, doi:10.3390/cancers15164125_

Round 1
Reviewer 1 Report
The authors reported the results of a review article with the aim of examining melanoma's phase III adjuvant treatment trials, focusing on the latest updates and reviewing the role of biomarkers in potentially individualizing adjuvant therapy and summarizing the main limitations and future direction of adjuvant therapy options for high-risk melanoma. The manuscript is interesting and well-written. However, I have some suggestions.
My comments:
- Introduction: treatment guidelines should be briefly summarized
- Material and Methods: this section is mandatory to summarize how you performed literature research
- Other treatment should be considered (doi:10.1093/annonc/mdz411)
- Discussion section is too brief. You should include your opinion on the use of adjuvant treatment and its possible use in Stage II melanoma
Author Response
To: Reviewer 1,
Journal of Cancers
Resubmission Date: August 10th, 2023
Dear Valued Reviewer,
Thank you for considering our manuscript “Adjuvant Therapy for High-risk Melanoma: An In-depth Examination of the State of the Field’ ID (cancers- 2557298) for review. We are grateful for the time and effort you have dedicated to providing valuable feedback on our manuscript. We are grateful for your comments on our paper. We have been able to incorporate changes to reflect your suggestions. We have highlighted the changes within the manuscript in track changes.
Here is a point-by-point response to your comments and concerns.
Comment 1: Introduction: treatment guidelines should be briefly summarized.
Response 1: The following paragraph was added to the introduction section in lines 75-90. “Systemic adjuvant therapies for high-risk melanoma, including interferon-α, have traditionally been used with limited uptake owing to limited efficacy and significant toxicity. However, recent years have seen the emergence of innovative therapies resulting from a deepening understanding of tumor biology and host immunology. These have included targeted therapies for BRAF mutant melanoma and ICI immunotherapies that have transformed the current standard of care with significant improvements in clinical outcomes, safety profiles and quality of life (QoL) [33]. Therefore, for patients with resected stage III or stage IV melanoma, current systemic adjuvant therapy options include ICI therapy with pembrolizumab or nivolumab and targeted therapy with dabrafenib-trametinib combination (for BRAF mutant melanoma). In addition, adjuvant pembrolizumab is an option for resected stages IIB and IIC melanoma where it has achieved regulatory approvals while nivolumab is currently under regulatory review for this indication.”
Comment 2: Material and Methods: This section must summarize how you performed literature research.
Response 2: The following paragraph was added to the introduction section in lines 90-93. “A literature search was performed through a comprehensive assessment of the published evidence on (MEDLINE with PubMed interface, date of the last search: May 31, 2023) in order to accommodate the inclusion of relevant articles for this review.”
Comment 3: Other treatments should be considered (doi:10.1093/annonc/mdz411)
Response 3: We agree with the reviewer and have reviewed all approved adjuvant therapeutic options with known clinical efficacy, including interferon-α and adjuvant radiation therapy.
Comment 4: The discussion section is too brief. You should include your opinion on adjuvant treatment and its possible use in Stage II melanoma.
Response 4: We have edited the manuscript to clarify the extent of the Discussion section. The title “Discussion” was added at line 291, highlighting all the subsections included. The section numbers were also updated. Our expert opinion has been included throughout the discussion in relation to specific points including in the summary section and has been further updated discussing possible use in Stage IIB-C melanoma and Stage IIIA patients with SLN involvement of <1mm on lines 476-502.
Sincerely,
Ahmad A. Tarhini, M.D., Ph.D.
Senior Member, Cutaneous Oncology and Immunology
- Lee Moffitt Cancer Center and Research Institute
10920 McKinley Dr. Tampa, FL 33612
Phone: 813-745-8581
Email: Ahmad.Tarhini@moffitt.org

Reviewer 2 Report
The review is very well written and comprehensive. Tables are accurate and the figure is nice.
I suggest to add a deep discussion on the safety profile of these drugs, with particular reference to the immune mediated adverse events such as the reactivation of pre-existing immune disease (cite the recent MA: PMID: 33314269)
Author Response
To: Reviewer 2,
Journal of Cancers
Resubmission Date: August 10th, 2023
Dear Valued Reviewer,
Thank you for considering our manuscript “Adjuvant Therapy for High-risk Melanoma: An In-depth Examination of the State of the Field’ ID (cancers- 2557298) for review. We appreciate the time and effort you have dedicated to providing valuable feedback on our manuscript. We are grateful for your comments on our paper. We have been able to incorporate changes to reflect your suggestions. We have highlighted the changes within the manuscript in track changes.
Here is a point-by-point response to your comments and concerns.
Comment 1: I suggest adding a deep discussion on the safety profile of these drugs, with particular reference to immune-mediated adverse events such as the reactivation of pre-existing immune disease (cite the recent MA: PMID: 33314269).
Response 1: The following paragraph was added under the section of Adjuvant Therapy Toxicity, starting on line 313, and the relevant references have been added as noted below.
“Since the anti-tumor efficacy of ICIs promotes various immune-related adverse events (irAEs), the paucity of knowledge regarding the safety of ICIs in patients with cancer and preexisting autoimmune disease represents a significant challenge. This is mainly due to the fact that patients with a history of symptomatic autoimmunity were excluded from clinical trials testing ICIs [74]. This question has prompted retrospective research efforts that examine the safety and tolerability of ICIs in patients with autoimmune diseases. For example, according to a recent [74], which included patients with multiple malignancies (84% were patients with melanoma) with active or inactive autoimmune diseases, three-quarters reported an aggravation of preexisting autoimmune diseases, the occurrence of new irAEs or both. Moreover, preexisting autoimmunity flares and irAEs in patients with autoimmune disease on ICIs can often be managed without the need for treatment discontinuation, while some adverse events can be severe and even fatal. In one of the largest series [75], which included treatment with ipilimumab in 30 patients with melanoma and preexisting autoimmune disease, it was found that around a quarter of patients experienced flares of the underlying autoimmunity and one third experienced new irAEs, including an event of death in one patient. Subsequently, the same group recently published a retrospective review of 52 patients treated with an anti-PD-1 antibody, with more than half (54%) having previously received ipilimumab. Specifically, underlying autoimmune diseases flared in 38% of patients, de novo irAEs occurred in 29%, and ICIs were discontinued in 12%. As expected, those with active preexisting autoimmune diseases experienced more flares than those with inactive diseases [76]. Finally, a more recent systematic review and meta-analysis [77] specifically looked at patients with pre-existing inflammatory bowel disease (IBD), and approximately 40% of the examined patients with melanoma experienced IBD relapse with ICIs, with most relapsing patients requiring corticosteroids and one third requiring biologics such as TNF blockers.
- Abdel-Wahab N, Shah M, Lopez-Olivo MA, Suarez-Almazor ME. Use of Immune Checkpoint Inhibitors in the Treatment of Patients With Cancer and Preexisting Autoimmune Disease: A Systematic Review. Ann Intern Med. 2018;168(2):121-30.
- Johnson DB, Sullivan RJ, Ott PA, Carlino MS, Khushalani NI, Ye F, et al. Ipilimumab Therapy in Patients With Advanced Melanoma and Preexisting Autoimmune Disorders. JAMA Oncol. 2016;2(2):234-40.
- Menzies AM, Johnson DB, Ramanujam S, Atkinson VG, Wong ANM, Park JJ, et al. Anti-PD-1 therapy in patients with advanced melanoma and preexisting autoimmune disorders or major toxicity with ipilimumab. Ann Oncol. 2017;28(2):368-76.
- Meserve J, Facciorusso A, Holmer AK, Annese V, Sandborn WJ, Singh S. Systematic review with meta-analysis: safety and tolerability of immune checkpoint inhibitors in patients with pre-existing inflammatory bowel diseases. Aliment Pharmacol Ther. 2021;53(3):374-82.
Sincerely,
Ahmad A. Tarhini, M.D., Ph.D.
Senior Member, Cutaneous Oncology and Immunology
- Lee Moffitt Cancer Center and Research Institute
10920 McKinley Dr. Tampa, FL 33612
Phone: 813-745-8581
Email: Ahmad.Tarhini@moffitt.org

Round 2
Reviewer 1 Report
All the changes have been made. The manuscript is now suitable for publication as it has been improved.
Reviewer 2 Report
The revised version of the paper is OK. Thank you!